# Passaging Primary Human Bronchial Epithelia Reduces CFTR-Mediated Fluid Transport and Alters mRNA Expression

**DOI:** 10.3390/cells12070997

**Published:** 2023-03-24

**Authors:** Tongde Wu, Joe A. Wrennall, Hong Dang, Deborah L. Baines, Robert Tarran

**Affiliations:** 1Department of Cell Biology & Physiology, University of North Carolina, Chapel Hill, NC 27599, USA; 2Marsico Lung Institute, University of North Carolina, Chapel Hill, NC 27599, USA; 3Institute for Infection and Immunity, St George’s, University of London, Tooting, London SW17 0RE, UK

**Keywords:** airway surface liquid, tight junction, membrane transport, Ussing chamber, water permeability

## Abstract

Primary human bronchial epithelial cultures (HBECs) are used to study airway physiology, disease, and drug development. HBECs often replicate human airway physiology/pathophysiology. Indeed, in the search for cystic fibrosis (CF) transmembrane conductance regulator (CFTR) therapies, HBECs were seen as the “gold standard” in preclinical studies. However, HBECs are not without their limitations: they are non-immortalized and the requirement for human donors, especially those with rare genetic mutations, can make HBECs expensive and/or difficult to source. For these reasons, researchers may opt to expand HBECs by passaging. This practice is common, but to date, there has not been a robust analysis of the impact of expanding HBECs on their phenotype. Here, we used functional studies of airway surface liquid (ASL) homeostasis, epithelial barrier properties, and RNA-seq and Western blotting to investigate HBEC changes over two passage cycles. We found that passaging impaired CFTR-mediated ASL secretion and led to a reduction in the plasma membrane expression of the epithelial sodium channel (ENaC) and CFTR. Passaging also resulted in an increase in transepithelial resistance and a decrease in epithelial water permeability. We then looked for changes at the mRNA level and found that passaging significantly affected 323 genes, including genes involved in inflammation, cell growth, and extracellular matrix remodeling. Collectively, these data highlight the potential for HBEC expansion to impact research findings.

## 1. Introduction

Bronchial epithelia are pseudostratified and contain ciliated cells and goblet cells which work in concert with submucosal glands to secrete protective mucins [1]. The ciliated cells have ~200 cilia per cell which beat in unison to propel mucus out of the lung [2]. They comprise ~80% of the surface epithelia and also contain the machinery to secrete the ions and water required to hydrate mucus. In contrast, goblet cells are ~20% of the epithelia and secrete gel-forming mucins such as MUC5AC and MUC5B. These epithelia can be isolated post-mortem or post-transplant and are typically cultured for 4–8 weeks. When cultured on plastic and/or submerged, they de-differentiate into non-polarized cells. However, when cultured on semi-permeable inserts at the air–liquid interface (i.e., submerged serosally, but with minimal fluid on the mucosal surface), they maintain their polarity and express ciliated and goblet cells [3]. Indeed, previous studies have demonstrated that ciliated cells in cultured airway epithelia have similar numbers of cilia per cell as seen in human airways in vivo, and maintain the 80:20 ratio of ciliated/goblet cells seen in vivo [4].

Human bronchial epithelial cultures (HBECs) have been an important test bed for the study of airway physiology/pathophysiology and for pulmonary drug development. They exhibit similar ion transport characteristics as seen in vivo and have predicted differences in salt/water transport as seen in CF patients [5,6,7]. Moreover, they respond to inflammatory stimuli with increased cytokine secretion [8] and can also undergo airway remodeling and goblet cell metaplasia [9]. They have been extensively used in both academia and the pharmaceutical industry for testing compounds. For example, in lieu of a suitable animal model, Vertex used HBECs to study VX-770 and VX-809 before moving into pre-clinical development and subsequent CF patient clinical studies [10,11]. Similarly, HBECs have been used to study epithelial Na channel (ENaC) antagonists [12,13]. Primary bronchial epithelia are non-immortalized/terminally differentiated cells, and immortalization leads to a loss of polarization (i.e., the ciliated and goblet cells disappear) [14]. Although primary (non-passaged) HBECs are robust, they are only available when tissue is available. They are also typically expensive and may not be available in sufficient quantities for high-throughput screening and/or large-scale drug development studies. To circumvent these issues, researchers have frozen and expanded primary HBECs into passage 1 (P1), passage 2 (P2), and beyond. However, despite the large amount of research that has been performed on HBECs, the full impact of passage on HBEC phenotypes has been under-studied. Previous studies have identified increases in inflammatory mediators in serially passaged HBECs [15], and investigations into using conditionally reprogrammed HBECs to expand proliferative capacity have identified negative effects of passage on fluid and ion transport [16]. Therefore, we compared primary (P0), P1, and P2 HBECs without conditional reprogramming from the same donors for ion/fluid transport; Western blotting and RNAseq were used to assess changes at the gene level.

## 2. Materials and Methods

### 2.1. Human Bronchial Isolation and Cell Culture

Human excess donor lungs and excised recipient lungs were obtained at the time of lung transplantation from portions of main stem or lumbar bronchi and cells were harvested by enzymatic digestion as previously described under a protocol approved by the UNC Institutional Review Board [17]. Cells from 3 non-smoker donors with no prior history of chronic lung disease were cultured directly (primary/P0 HBECs) or grown on plastic in bronchial epithelial growth media (LHC basal medium [Life Technologies Carlsbad, CA, USA] supplemented with Bovine serum albumin [0.5 mg/mL, Sigma-Aldrich St. Louis, MO, USA], bovine pituitary extract [10 μg/mL, Sigma-Aldrich], Insulin [0.87 μM, Sigma-Aldrich], Transferrin [0.125 μM, Sigma-Aldrich], Hydrocortisone [0.21 μM, Sigma-Aldrich], Triiodothyronine [0.01 μM, Sigma-Aldrich], Epinephrine [2.7 μM, Sigma-Aldrich], epidermal growth factor [25 ng/mL, Invitrogen], retinoic acid, 5  ×  10^−8^ M, Sigma-Aldrich], Phosphorylethanolamine [0.5 μM, Sigma-Aldrich], Ethanolamine [0.5 μM, Sigma-Aldrich], zinc sulfate [3.0 μM, Sigma-Aldrich], Penicillin G sulfate [100 U/mL, Sigma-Aldrich], streptomycin sulfate [100 μg/mL, Sigma-Aldrich], Gentamicina [50 μg/mL, Sigma-Aldrich], and Amphotericin [0.25 μg/mL, Sigma-Aldrich]) for 1 and 2 passages before culture under air–liquid interface (ALI) conditions until full differentiation in media on culture inserts (6.5 mm diameter, polyester 0.4 μm pore size; Costar 3470, (Fisher Scientific, Pittsburgh, PA, USA). All cells were cultured for 28–35 days.

### 2.2. Airway Surface Liquid (ASL) Height Measurements

Cultures were washed three times with PBS mucosally, with the last wash extended for 30 min at 37 °C in a 5% CO_2_ incubator. After aspiration of all PBS, 0.5 mg/mL rhodamine–dextran (Life Technologies, 10 kDa, neutral) in 20 µL Ringer’s solution without glucose was added mucosally and incubated for 15 min at 37 °C/5% CO_2_. Excess dye was aspirated from the mucosal surface by Pasteur pipette, leaving residual dye that was ~7 µm in height; cultures were further incubated for 30 min in the incubator and then ASL measurements were taken. Cells were kept in Ringer’s solution during imaging with perfluorocarbon (FC-770, 3M Fluorinert^TM^) added mucosally to prevent evaporation of the ASL [18]. For live cell staining, cells were stained with 3 µM calcein-AM (Life Technologies) in PBS for 30 min at 37 °C/5% CO_2_, followed by three rapid PBS washes and further staining of ASL with rhodamine–dextran, as mentioned above. Images were taken on a Leica SP8 confocal microscope with a 63X glycerol immersion objective, as described previously [19]. ASL height was determined using ImageJ software (National Institutes of Health freeware).

### 2.3. Western Blotting

HBECs were washed with ice-cold PBS and lysed with buffer containing 25 mM Tris, 150 mM NaCl, 1% NP-40, 1% sodium deoxycholate pH 7.4, protease inhibitors (cOmplete, EDTA-free; Roche Applied Science) on ice for 15 min with occasional shaking. The lysate was centrifuged at 5000× *g* for 10 min at 4 °C and the supernatant was used for Western blotting. For the assessment of apical membrane proteins, apical membrane proteins were biotinylated, as previously described [20]. Polarized HBECs were washed three times with PBS supplemented with 1 mm MgCl_2_ and 1 mm CaCl2 (PBS^2+^). Sulpho-NHS-biotin (0.5 mg mL^−1^) in borate buffer (in mm: 85 NaCl, 4 KCl, 15 Na_2_B_4_O_7_, pH 9) was applied apically and incubated for 30 min with gentle agitation. PBS^2+^ supplemented with 10% (*v*/*v*) FBS was added to the basolateral bath to prevent the biotinylation of basolateral proteins. Unbound sulpho-NHS-biotin was quenched with PBS^2+^ supplemented with 10% (*v*/*v*) FBS. Cells were lysed in lysis buffer (0.4% sodium deoxycholate, 1% NP-40, 50 mm EGTA, 10 mm Tris-Cl, pH 7.4 and protease inhibitor (Roche), and the protein concentration was determined by bicinchoninic acid assay. Subsequently, 300 μg of total protein per sample was incubated overnight with 100 μL of neutravidin–agarose beads at 4 °C with agitation. Biotinylated proteins bound to beads were washed three times with lysis buffer and eluted in 30 μL of Laemmli buffer supplemented with 10% (*v*/*v*) β-mercaptoethanol, by first incubating them at room temperature for 10 min, followed by heating at 95 °C for another 10 min. For both apical and total protein experiments, 50 µg of protein was resolved using 4–15% Bio-Rad Mini-Protean TGX gels and transferred to PVDF membrane. Membranes were blocked with 5% non-fat dry milk in Tris-buffered saline with 0.2% Tween-20 for 2 h at room temperature, followed by incubation with primary antibody overnight at 4 °C in 5% non-fat dry milk in Tris-buffered saline with 0.2% Tween-20 (1:5000 596 CFTR antibody that recognizes aa1204-1211 on NBD2 domain, Cystic Fibrosis Foundation Therapeutics Inc. Bethesda, MD, USA). Immunostained proteins were visualized with HRP conjugated secondary antibodies (Jackson ImmunoResearch Laboratories Inc. West Grove, PA, USA) and developed using BioRad Clarity Western ECL substrate using a ChemiDoc^TM^ MP Imaging system (Bio-Rad, Hercules, CA, USA). Blots were stripped and reprobed for GAPDH (anti GAPDH antibody, Cell Signaling Technology, Danvers, MA, USA) (1:2000).

### 2.4. Measurement of Transepithelial Water Flow

HBECs were bilaterally loaded with 3 μM calcein-AM (Invitrogen, Waltham, MA, USA) for 30 min at 37 °C. Calcein-loaded HBECs were observed by XZ confocal microscopy. Cultures were placed in isotonic Ringer’s solution containing 0.1 mg/mL rhodamine–dextran in an Attofluor chamber (ThermoFisher, Waltham, MA, USA) on the confocal microscope. Transepithelial water flow was initiated by the mucosal addition of 200 μL of hyperosmotic solution (Ringer plus 75 mM NaCl); then cell height and serosal bath rhodamine–dextran fluorescence intensity were tracked over 48 s by obtaining repeat XZ confocal images (1 image every 2 s), as described previously [20]. Cell height and serosal rhodamine–dextran were then analyzed using NIH Image J software.

### 2.5. Transepithelial Electrical Resistance Measurements

Transepithelial electrical resistance was measured using an electrovoltometer (EVOM) (World Precision Instruments) using an STX2 electrode, as previously described [21]. Here, 300 μL PBS was added mucosally for 15 min before study. Resistance was zeroed using a blank culture and transepithelial resistance was read.

### 2.6. RNAseq Analysis

Total mRNA was extracted from 4-week old, ALI HBECs using RNaEasy reagent (Qiagen, Hilden, Germany). Raw sequence reads (paired-ends of 150 base pairs), were mapped to the current reference genome (GRC38) and primary gene annotation from Gencode v40 (https://www.gencodegenes.org (accessed on 21 February 2023)), using the aligner, STAR (https://github.com/alexdobin/STAR (accessed on 21 February 2023)). Gene expression was quantified using a StringTie transcriptome assembler (http://ccb.jhu.edu/software/stringtie/index.shtml (accessed on 21 February 2023)). Differential gene expression was analyzed using linear models in the Bioconductor package, *variance Partition*. Genes were filtered by minimal expression: total counts across samples >100 and normalized using Voom [22]. Data were interpreted and presented using volcano plot and gene set enrichment analysis (http://GSEA-MSigDB.org (accessed on 21 February 2023)) [23,24] using the c5.go.bp.v2022.1.Hs.symbols.gmt database of GO Biological Process ontology gene sets available on the molecular signatures data base [25]. Functional relationships between enriched gene sets were mapped using the EnrichmentMap plugin of Cytoscope V3.9.1. Leading edge analysis was applied to the 50 most enriched GO BP gene sets to identify the genes with the highest impact on gene set enrichment.

### 2.7. Statistical Analysis

Statistical analysis was performed using Graph Pad Prism and Graph Pad Instat. Data were analyzed using Freidman tests (non-parametric matched ANOVA) with Dunn’s multiple comparisons post-test. ASL secretion data were fitted with linear equations. For the water flux experiments, both serosal bath intensity and cell height data were fitted with a one-phase exponential equation (Y = (Y0 − Plateau) * exp(−K * X) + Plateau) using GraphPad Prism. Fitted graphical data were analyzed with the extra-sum-of-squares F test.

## 3. Results and Discussion

We have previously used mostly primary, but also P1 HBECs, for our epithelial transport studies, and we have not tested whether passage number affects phenotype. To evaluate whether vectoral fluid transport was impaired by passage, we measured ASL height by confocal microscopy over time. ASL homeostasis is an integration of active anion secretion vs. cation absorption, along with passive movement of the corresponding counter ion and water [26]. HBECs typically absorb excess ASL in a Na^+^/ENaC-led manner, after which point, soluble ENaC inhibitors (e.g., the short palate lung and nasal epithelial clone 1 (SPLUNC1) accumulate, leading to ENaC inhibition and the predominance of cAMP-led anion secretion via CFTR [27]. To track ASL height, we added a bolus of test solution (20 μL of PBS). Regardless of passage, HBECs absorbed ASL similarly over the initial 8 h (Figure 1A). However, by 24 h, P2 cultures had a significantly lower steady-state ASL height compared with P0 cultures (Figure 1A,B). At steady state (i.e., 24 h after PBS addition), we then added adenosine (ADO) mucosally as a dry powder in perfluorocarbon to stimulate CFTR-mediated ASL secretion, as described previously [28]. P0 cultures robustly secreted ASL in response to ADO; however, ADO-induced secretion was reduced in P1 cultures and essentially abolished in P2 cultures (Figure 1C,D), suggesting that passage had a greater impact on anion secretion than on Na^+^ absorption.

To further study this phenomenon, we performed surface biotinylation followed by Western blot on primary and passaged HBECs. αENaC was moderately decreased in the apical membrane (*p* = 0.08); however, total αENaC significantly decreased over 2 passages (Figure 2A,B,D). We did not detect any change in total CFTR (Figure 2A,C,E). However, consistent with the decrease in ADO-mediated anion secretion (Figure 1), we observed a significant decrease in CFTR protein at the apical plasma membrane. These changes suggested that the decrease in apical membrane protein abundance may involve altered protein trafficking rather than a decrease in total CFTR. The apparent trend towards decreased apical αENaC levels did not correlate with the observed changes in ASL absorption, because reduced ENaC would be predicted to increase ASL height. However, ENaC was likely working under basal conditions and may not have been maximally stimulated since exogenous proteases that can cleave and activate ENaC were not added to the system [29]. In contrast, passage seemed to affect both ASL secretion and CFTR expression, suggesting that CFTR-mediated anion secretion was more affected by passage than ENaC. Indeed, we have previously observed that the inhibition of CFTR or basolateral Cl^−^ uptake also results in a reduction in steady-state ASL height [30]. Thus, we speculate that the 24 h decrease in ASL height seen in P2 HBECs (Figure 1B) may have been caused by the lack of CFTR.

Next, we measured the impact of passage on HBEC water permeability. As previously described, we exposed HBECs to a hypertonic saline challenge (450 mOsm) and then measured the rate of cell shrinkage as a marker of transapical water permeability [20]. We also measured the concentration of an impermeable dye in the serosal compartment (rhodamine conjugated to 10 kDa dextran) as a marker of transepithelial water permeability. Passage from P0 to P2 resulted in a significant decrease in cell shrinkage, indicating that HBEC passage significantly decreased transapical water permeability (Figure 3A,B). The concentration of rhodamine trended towards being reduced following the osmotic challenge (Figure 3C,D). Thus, these data predict that fluid transport, a key feature of epithelial function, would be diminished with passage.

We also found that passage number significantly increased the transepithelial electrical resistance, with P2s exhibiting more than double the resistance of P0s (Figure 3E). Notably, airway epithelia are classed as “leaky” epithelia, meaning that ions and some uncharged solutes can pass paracellularly [31]. Indeed, freshly isolated airway epithelia have a transepithelial resistance of ~120 Ω.cm^2^ and, in keeping with our previous studies, we found P0 HBEC resistance to be around 300 Ω.cm^2^, which is still sufficiently low to allow for paracellular ion movement. However, increasing the transepithelial resistance could impede the ability of the counter ion to move paracellularly, limiting the ability of the cultures to isotonically move salt and water via CFTR-mediated secretion or ENaC-led absorption. Taken together, it is likely that the reduction in water permeability and transepithelial resistance, especially that seen in P2 cultures, contributed to the reduced adenosine-mediated ASL secretion (Figure 1).

Finally, we performed RNAseq on P0 and P2 cultures to gain a broader understanding of the effect of passage on gene expression. After normalizing by quartiles, we corrected for multiple comparisons (see the Methods and Methods), where an adjusted *p* value (correcting for multiple tests) of 0.05 was deemed significant. The entire dataset is shown in Appendix A. We found that the majority of transcripts were significantly upregulated (191 upregulated vs. 132 down-regulated; Figure 4A). In line with our observation of broad transcriptional changes, several transcriptional regulators were identified in the list of the most significantly altered genes, such as insulin-like growth factor 2 mRNA-binding protein 3 (IGF2BP3), E1A-binding protein p400 (EP400), period circadian protein homolog 2 (PER2), and immediate early-response gene 5 protein (IER5). Ubiquitin D (UBD), a protein reported to mediate mitotic non-disjunction and chromosome instability in long-term in vitro culture 29, was upregulated more than 14-fold. As might be expected, regulators of the cell cycle, cell death, and cell–extracellular matrix (ECM) interactions dominated the list (Appendix A).

To better understand the implications of these changes, gene set enrichment analysis was performed using a curated list of gene sets based on known biological processes. Of the 4291 gene sets analyzed, 1603 were significantly enriched at a nominal *p* value of <1%. An annotated enrichment map is shown in Figure 4B. In line with Reeves et al. (2018), the most numerous changes were observed in inflammatory signaling pathways. Pathways related to apoptosis, wound response, and misfolded protein response were also well represented, suggesting cell stress. The top 50 enriched gene sets are shown in Table 1. Leading edge analysis was used to identify the most impactful genes within the gene sets of this table (Figure 4C). In line with the enrichment map, these genes consisted primarily of inflammatory mediators such as interleukins, chemokines, and genes associated with antigen presentation. The list also included integrins and metalloproteases, which were indicative of extracellular matrix remodeling. The pro-inflammatory transcription factors GATA3 [32] and STAT1 [33] were also enriched, which may be driving this phenotype. Multiple members of the placenta-derived growth factor (PDGF) pathway were also enriched, suggesting that passage alters cellular proliferation. For example, in airway epithelia, PDGF signaling promotes epithelial-to-mesenchymal transition [34]. Additionally, PDGF signaling can activate STAT1 [35], linking PDGF signaling to many of the most impactful inflammatory mediators associated with passage. This, together with the broad enrichment of pro-inflammatory signaling, may indicate that serially passaged HBEC cultures may alter gene expression as an adaptation to the cell culture environment.

Helman et al. previously demonstrated that culture media and substrate had a significant effect on ENaC expression in A6 epithelial cells [36]. In our study, we did not examine the impact of culture substrate or culture media on ion/fluid transport and mRNA expression. However, it is likely that these factors also play a role. Due to the high cost of primary HBECs, and their inconvenience (i.e., they are not always readily available), passaged cultures are extensively used for research in asthma, cystic fibrosis, smoking/COPD, etc. It has implicitly been assumed that the passaging of primary bronchial epithelia is an acceptable way to circumvent the availability issue. However, to the best of our knowledge, this is the first study to assess the impact of passage on HBEC function and mRNA expression. Taken together, these data indicate that passaging exerts a significant effect on bronchial epithelia. This does not mean that passaging should be avoided. However, researchers should ensure that the HBECs maintain the function of interest at any given passage number.

With passage, we observed a decrease in all functions studied, while at the mRNA level, we saw both upregulation and downregulation, that did not always correlate with the functional changes. For example, the aquaporin AQP4 was downregulated at the mRNA level, which correlated with decreased fluid transport (Figure 1 and Figure 3). In contrast, although we observed a decrease in transepithelial resistance, we did not detect a change in the expression of tight junction genes such as claudins. Similarly, we detected no change in CFTR at the gene level, but CFTR was diminished in the apical membrane and CFTR-mediated fluid secretion was diminished. Thus, changes at the mRNA level may not fully reflect changes in protein expression, localization, and/or functional changes. However, in general, the large numbers of altered transcripts seen with passage were consistent with our (admittedly not extensive) series of physiological and biochemical assays.

## 4. Conclusions

In conclusion, our study demonstrates that passaging HBECs alters gene expression and changes functions associated with ASL volume homeostasis. However, passaged, non-immortalized HBECs are likely closer to the in vivo situation than immortalized airway epithelia that do not produce cilia (e.g., A549 or CALU3 airway cell lines). Moreover, HBECs are non-clonal and contain several cell types including ciliated cells, goblet cells, basal cells and ionocytes. Thus, in balance, passaging remains a useful tool to extend the life of these cultures.

## Figures and Tables

**Figure 1 cells-12-00997-f001:**
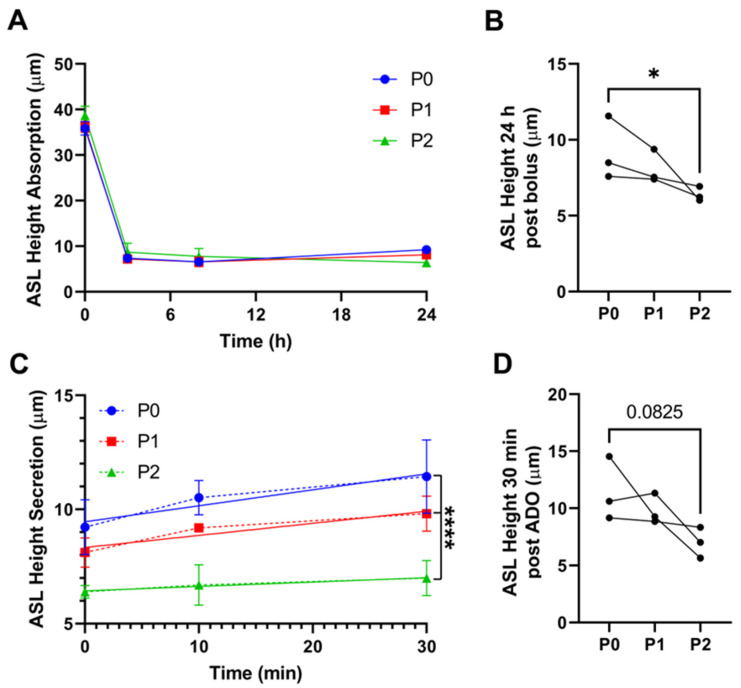
Passaging HBECs alters epithelial absorption and secretion dynamics. (**A**) ASL height at various timepoints after ASL was labeled with a bolus of rodamine–dextran. (**B**) ASL height 24 h after the bolus was added. (**C**) ASL height after adenosine (ADO) was added to stimulate CFTR-mediated secretion. (**D**) ASL height 30 min after adenosine was added * = *p* < 0.05, Friedman test. **** = *p* < 0.001 intercepts between fits; *n* = 3.

**Figure 2 cells-12-00997-f002:**
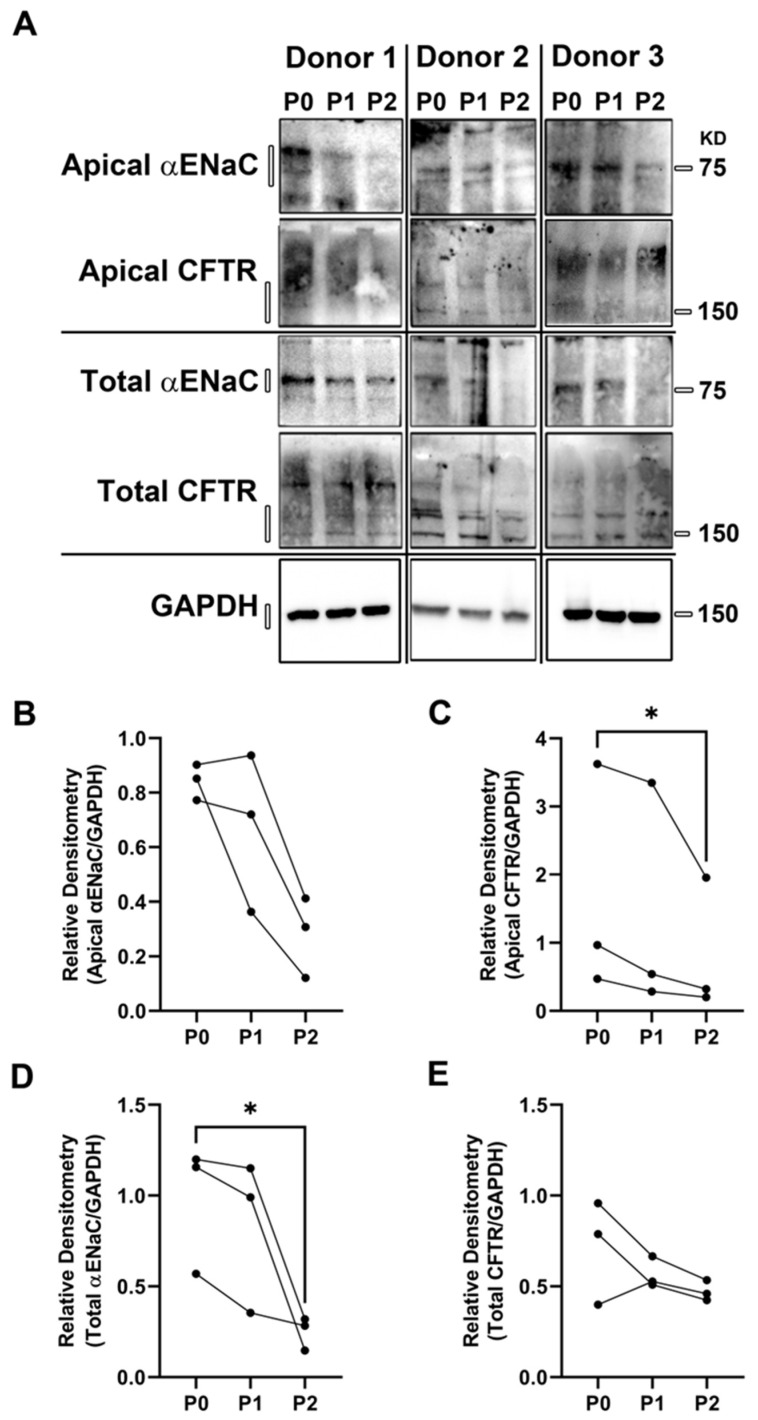
Passaging HBECs alters the apical membrane insertion of αENaC and CFTR. (**A**) Western blots of differentiated HBEC ALI cultures probing for αENaC and CFTR in the apical membrane or whole cell. Densitometry analysis of apical αENaC (**B**), apical CFTR (**C**), total αENaC (**D**), and total CFTR (**E**) are shown normalized to GAPDH. * = *p* < 0.05, *n* = 3, Friedman test.

**Figure 3 cells-12-00997-f003:**
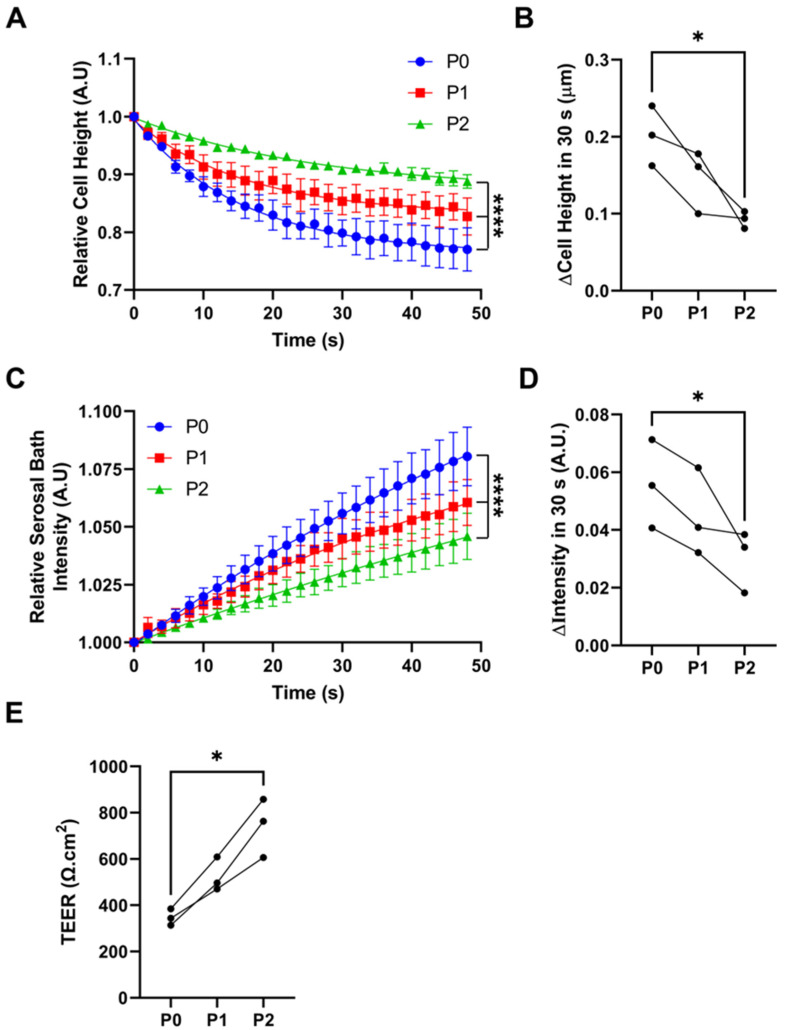
Passaging HBECs alters cell volume regulation. (**A**) Cell height in response to mucosal addition of 200 mL of hyperosmotic solution. (**B**) Summary cell height data. (**C**) Serosal bath intensity over time after the serosal addition of a bolus of rodamine–dextran-labeled saline. (**D**) Summary serosal bath intensity data. (**E**) Transepithelial resistance (T.E.R.) of differentiated HBEC ALI cultures at different passage numbers. *n* = 3, * = *p* < 0.05, Friedman test. **** *p* < 0.001 extra-sum-of-squares F test.

**Figure 4 cells-12-00997-f004:**
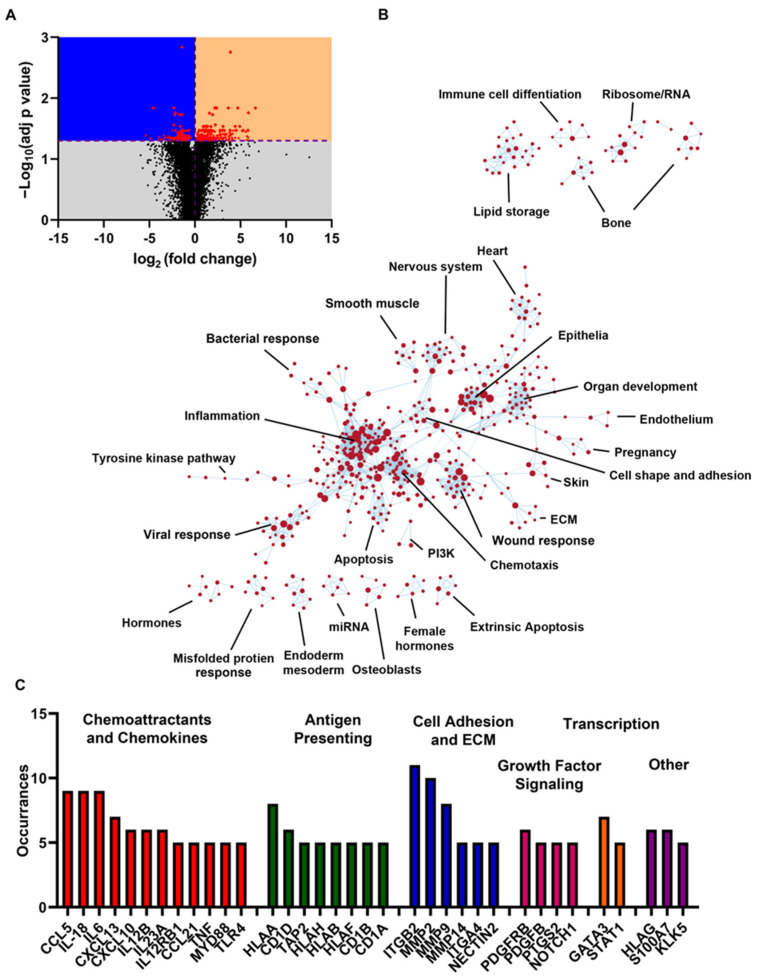
Passaging HBEC alters gene expression. (**A**) Volcano plot of RNAseq data comparing P0 and P2 HBECs. Purple dashed lines represent *p* = 0.05 and log_2_ (fold change) = 0. Black dots are genes with non-significant changes. Red dots are genes with significant changes. (**B**) String analysis of enriched gene sets (c5.go.bp.v2022.1.Hs.symbols.gmt database of GO Biological Process ontology gene sets. (**C**) Most impactful genes identified by leading edge analysis of significantly enriched gene sets shown in panel B. Occurrences indicates the number of significantly enriched gene sets containing the indicated gene.

**Table 1 cells-12-00997-t001:** GO pathways identified gene set enrichment analysis. RNAseq data were compared between P0 and P2 HBECs using GSEA_4.3.2 and ranked based on normalized enrichment score (NES). FDR, false discovery rate.

Category	GO BP Pathways	SIZE	NES	NOM *p*-Val	FDR q-Val
Inflammation	NEGATIVE REGULATION OF VIRAL GENOME REPLICATION	57	2.5107388	0	0.002015
	ANTIMICROBIAL HUMORAL IMMUNE RESPONSE MEDIATED BY ANTIMICROBIAL PEPTIDE	81	2.461054	0	0.002411
	POSITIVE REGULATION OF INTERLEUKIN 8 PRODUCTION	63	2.4569626	0	0.002233
	POSITIVE REGULATION OF T-CELL-MEDIATED IMMUNITY	59	2.4206753	0	0.005766
	RESPONSE TO INTERFERON ALPHA	20	2.4022737	0	0.005885
	POSITIVE REGULATION OF T CELL MIGRATION	35	2.3621593	0	0.010333
	POSITIVE REGULATION OF INTERLEUKIN 17 PRODUCTION	27	2.3206577	0	0.013589
	RESPONSE TO INTERFERON BETA	31	2.3061914	0	0.011949
	GOBP-POSITIVE REGULATION OF T-CELL-MEDIATED CYTOTOXICITY	31	2.3033602	0	0.011539
	CHRONIC INFLAMMATORY RESPONSE	19	2.299529	0	0.011256
	MACROPHAGE CYTOKINE PRODUCTION	37	2.2872465	0	0.01198
	POSITIVE REGULATION OF T CELL CYTOKINE PRODUCTION	26	2.2670827	0	0.01348
	ANTIGEN PROCESSING AND PRESENTATION OF ENDOGENOUS ANTIGEN	26	2.2438183	0	0.014412
	POSITIVE REGULATION OF T CELL PROLIFERATION	106	2.2167554	0	0.01559
	EOSINOPHIL CHEMOTAXIS	18	2.1964965	0.001727116	0.01663
	REGULATION OF T-CELL-MEDIATED IMMUNITY	88	2.1958168	0	0.016168
	POSITIVE REGULATION OF LYMPHOCYTE MIGRATION	42	2.189307	0	0.015607
	REGULATION OF T-CELL-MEDIATED CYTOTOXICITY	41	2.1841776	0	0.016023
	ANTIMICROBIAL HUMORAL RESPONSE	126	2.1838183	0	0.015743
	T CELL CHEMOTAXIS	28	2.174991	0	0.015316
	NEUTROPHIL CHEMOTAXIS	104	2.1733332	0	0.015079
Cell cycle/growth/differentiation	POSITIVE REGULATION OF SMOOTH MUSCLE CELL PROLIFERATION	98	2.3246183	0	0.014094
	RESPONSE TO PLATELET-DERIVED GROWTH FACTOR	26	2.3077137	0	0.012514
	REGULATION OF SUBSTRATE ADHESION DEPENDENT CELL SPREADING	61	2.2591238	0	0.013544
	GOBP RESPONSE TO HYPEROXIA	20	2.2540278	0	0.014027
	GOBP ENDODERMAL CELL DIFFERENTIATION	49	2.2402594	0	0.014091
	RESPONSE TO INCREASED OXYGEN LEVELS	27	2.2108958	0	0.015983
	POSITIVE REGULATION OF SUBSTRATE ADHESION DEPENDENT CELL SPREADING	44	2.1757176	0	0.015631
	MESENCHYMAL-TO-EPITHELIAL-TRANSITION	20	2.158714	0	0.016202
	KERATINOCYTE DIFFERENTIATION	169	2.1640036	0	0.015958
	GLIAL CELL PROLIFERATION	54	2.200997	0	0.01634
	CELL PROLIFERATION INVOLVED IN KIDNEY DEVELOPMENT	23	2.1467042	0	0.0169
	EPITHELIAL CELL DIFFERENTIATION INVOLVED IN KIDNEY DEVELOPMENT	48	2.1437025	0	0.017103
	MYELOID DENDRITIC CELL DIFFERENTIATION	20	2.143119	0	0.016788
ECM	COLLAGEN CATABOLIC PROCESS	45	2.4721029	0	0.003573
	EXTRACELLULAR MATRIX DISASSEMBLY	64	2.3468978	0	0.011455
Signaling	REGULATION OF RECEPTOR RECYCLING	22	2.2108297	0.001628664	0.01545
	RECEPTOR RECYCLING	33	2.1946115	0	0.015931
	SEMAPHORIN PLEXIN SIGNALING PATHWAY	44	2.1574514	0	0.016083
	CELLULAR RESPONSE TO MECHANICAL STIMULUS	73	2.154527	0	0.016141
Other	KERATINIZATION	83	2.5548687	0	0.001339
	EMBRYO IMPLANTATION	59	2.461652	0	0.003013
	ENDODERM FORMATION	58	2.3102279	0	0.013374
	BRANCHING INVOLVED IN BLOOD VESSEL MORPHOGENESIS	36	2.2791824	0	0.012697
	OVULATION	22	2.2394168	0	0.01369
	ADHESION OF SYMBIONT TO HOST	17	2.2286227	0.001718213	0.015132
	DECIDUALIZATION	25	2.2207658	0	0.015569
	POSITIVE REGULATION OF BONE RESORPTION	18	2.1918597	0.001779359	0.015708
	ENDODERM DEVELOPMENT	83	2.1805403	0	0.015548
	GLOMERULAR MESANGIUM DEVELOPMENT	18	2.1641705	0.001724138	0.016329

## Data Availability

Data is contained within the article or Appendix A.

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
