# Peer review of "Passaging Primary Human Bronchial Epithelia Reduces CFTR-Mediated Fluid Transport and Alters mRNA Expression"

_cells, 2023, doi:10.3390/cells12070997_

Round 1
Reviewer 1 Report
Within the current manuscript, Wu et al. report functional changes in primary human bronchial epithelial cultures (HBEC) over 2 passage cycles. HBEC are widely used to study human airway physiology and disease, replicating many functional features of the intact airway epithelium. As rightly indicated by the authors, a main limitation of this excellent model is the availability of sufficient cell numbers / donors. Therefore, various protocols have been established in the past to expand HBECs by passaging, often assuming or relying on perpetuation of the functional characteristics of the primary cells over few passages. This timely study now highlights significant changes in epithelial function over 2 passage cycles. This is an important finding to sensibilize researchers using this model and taking into account these changes when interpreting their findings. Overall, the study is well performed, utilizing established assays to characterize epithelial function and reporting transcriptional changes identified via RNA-seq and Western blotting. Some concerns, however, remain over data representation and statistical analysis. These need to be addressed. In addition, I find that limiting the analysis to changes in epithelial barrier and transport properties, prevents a more holistic depiction of the overall effect of cell culture / passage on HBECs. A more complete analysis of changes including changes in cellular composition or mucus secretion would substantially improve the current study.
Major:
- The quality of the Western blots (Fig 2) is very poor. First of all, specificity of the signals, in particular the CFTR signal, is not clear. Some blots show 1 other 2 bands. This questions the accuracy of the quantification. It also seems that the quantification of total ENaC expression does not reflect the WB signal. At P2 there is almost no signal visible (at least in donor 2 and 3) however, quantification indicates hardly any reduction in expression. Therefor the WB images should be replaced with higher quality blots, ideally showing the entire length of the probed membranes (instead of single bands). Please also display data in Fig 2B, C as non-normalized values
- The authors state that cells were obtained from 4 donors. However, some graphs display many more data points. Please indicate whether n represents donors or cultures. If n represents cultures (not donors), data need to be weighed to avoid over-representation of individual donors (e.g use means for each donor).
- Specification of statistical analysis is somewhat unclear. The legend of Fig 2 states that ANOVA was used for comparing multiple samples whereas the methods indicate the use of Kruskal-Wallis test. These are fundamentally different analyses with regards to the prerequisite of a Gaussian distribution of values. Please make sure to use and indicate the appropriate statistical method. Data in fig
- A serious limitation for comparing the data with observations in other labs is, that using passages (i.e. P0, P1, P2) as a measure of cell culture time is somewhat inappropriate, it does not take into account cell proliferation per se. It would be more accurate to compare the changes to the number of cell doublings.
- If available, I would appreciate the inclusion of information on changes in cellular composition or mucus secretion over the 2-passage time course. This could substantially contribute to a more holistic evaluation of the changes in HBECs with passaging.
Minor:
- Generally, data representation in bar graphs does not accurately present a time course (as it is over increasing passages). Maybe the authors can change representation to line plots (with symbols and error bars).
- Fig 1: y-axis labels state (μM) instead of (μm) for ASL height
- Fig 3E, TER is represented in Ohm, when TER is usually presented as Ohm*cm2.
- Table 1: I´m sure the value “Sep-63” in the Count of network column resulted from some format conversion mishap.
Author Response
Reviewer 1
Within the current manuscript, Wu et al. report functional changes in primary human bronchial epithelial cultures (HBEC) over 2 passage cycles. HBEC are widely used to study human airway physiology and disease, replicating many functional features of the intact airway epithelium. As rightly indicated by the authors, a main limitation of this excellent model is the availability of sufficient cell numbers / donors. Therefore, various protocols have been established in the past to expand HBECs by passaging, often assuming or relying on perpetuation of the functional characteristics of the primary cells over few passages. This timely study now highlights significant changes in epithelial function over 2 passage cycles. This is an important finding to sensibilize researchers using this model and taking into account these changes when interpreting their findings. Overall, the study is well performed, utilizing established assays to characterize epithelial function and reporting transcriptional changes identified via RNA-seq and Western blotting. Some concerns, however, remain over data representation and statistical analysis. These need to be addressed. In addition, I find that limiting the analysis to changes in epithelial barrier and transport properties, prevents a more holistic depiction of the overall effect of cell culture / passage on HBECs. A more complete analysis of changes including changes in cellular composition or mucus secretion would substantially improve the current study.
Major:
- The quality of the Western blots (Fig 2) is very poor. First of all, specificity of the signals, in particular the CFTR signal, is not clear. Some blots show 1 other 2 bands. This questions the accuracy of the quantification. It also seems that the quantification of total ENaC expression does not reflect the WB signal. At P2 there is almost no signal visible (at least in donor 2 and 3) however, quantification indicates hardly any reduction in expression. Therefore the WB images should be replaced with higher quality blots, ideally showing the entire length of the probed membranes (instead of single bands). Please also display data in Fig 2B, C as non-normalized values
[Response] We appreciate that the quality of the western blots presented here may be suboptimal. Unfortunately, CFTR is expressed at the limit of detection in our HBEC culture system. Thus, due to the use of HBECs in this study and the difficulty in detecting endogenous low abundance proteins, clean, high contrast blots are technically challenging. These difficulties have also been encountered by others studying CFTR (e.g. PMID: 32457197). Importantly, we believe that these changes in CFTR protein levels are consistent with our functional data which demonstrate impairments in fluid transport with passage. Thus, these blots are valid and highlight an important impact of passage on the expression of key ion channels. We hope that the quality of our western blots can be re-evaluated in the context of the technical challenges of the culture model used.
In line with the reviewer’s comments, we have expanded the blots to display more of the lanes being quantified. Size markers have also been added and the summary data has been updated to remove normalization to passage 0 for each donor.
- The authors state that cells were obtained from 4 donors. However, some graphs display many more data points. Please indicate whether n represents donors or cultures. If n represents cultures (not donors), data need to be weighed to avoid over-representation of individual donors (e.g use means for each donor).
[Response] We thank the reviewer for highlighting an error in our methods. 3 donors were used throughout the study. For each experiment, the 3 donors are equally weighted (i.e. the same number of cultures were included for each donor). In line with the reviewer’s suggestion, we have updated our figures and statistics to present 1 data point per donor. RNAseq analysis remains as 2 cultures per donor from 3 donors.
- Specification of statistical analysis is somewhat unclear. The legend of Fig 2 states that ANOVA was used for comparing multiple samples whereas the methods indicate the use of Kruskal-Wallis test. These are fundamentally different analyses with regards to the prerequisite of a Gaussian distribution of values. Please make sure to use and indicate the appropriate statistical method. Data in fig
[Response] We thank the reviewer for highlighting this discrepancy. For all comparisons (excluding RNAseq) we have used Freidman tests (non-parametric matched ANOVA). We have updated the manuscript accordingly.
- A serious limitation for comparing the data with observations in other labs is, that using passages (i.e. P0, P1, P2) as a measure of cell culture time is somewhat inappropriate, it does not take into account cell proliferation per se. It would be more accurate to compare the changes to the number of cell doublings.
[Response] We thank the reviewer for their suggestion. We agree that the use of cell doublings as a metric for culture times would be an excellent approach with a higher degree of resolution than passage number. Unfortunately, this data is not available to us and we do not believe that there is a viable approach to accurately calculate these values from our data. In our experience, protocols regarding HBEC cultures often rely on passage number for tracking expansion instead of the use of raw cell counts. For this reason, we believe that the use of passage number is a fair metric with real world application. Importantly, the central thesis of our manuscript is that the expansion of HBEC samples has meaningful impacts on HBEC biology that should be considered during experimental design. While the use of cell doublings may provide more detailed understanding of the degree of expansion possible before such changes occur, we believe that the use of passage number is sufficient to demonstrate that changes occur.
- If available, I would appreciate the inclusion of information on changes in cellular composition or mucus secretion over the 2-passage time course. This could substantially contribute to a more holistic evaluation of the changes in HBECs with passaging.
[Response] The reviewer has highlighted useful further experiments that could be performed to increase the scope of our characterization of HBECs during expiation. Unfortunately, we did not collect this data during our experiments. However, we shall certainly consider this metric in the future.
Minor:
- Generally, data representation in bar graphs does not accurately present a time course (as it is over increasing passages). Maybe the authors can change representation to line plots (with symbols and error bars).
[Response] We thank the reviewer for their suggestion. We agree that line plots better present time-course data and have updated our figures accordingly.
- Fig 1: y-axis labels state (μM) instead of (μm) for ASL height
[Response] We thank the reviewer for identifying this error. It has been corrected.
- Fig 3E, TER is represented in Ohm, when TER is usually presented as Ohm*cm2.
[Response] We thank the reviewer for identifying this error. It has been corrected.
- Table 1: I´m sure the value “Sep-63” in the Count of network column resulted from some format conversion mishap.
[Response] We thank the reviewer for identifying this error. We have updated our tables to reflect our improved RNAseq analysis and resolved this error.
Reviewer 2 Report
The manuscript from Tongde Wu et al. reports functional and transcriptomic alterations in passaged primary human bronchial epithelial cells. The findings are important for the research community exploiting airway epithelial cell cultures. The manuscript is well-written but is often brief. The experimental design could have been more ambitious to strengthen the conclusions. It seems better suited as a short communication. In addition, why is the chosen section “cellular immunology”?
Major comment:
1. The graphical abstract is misleading for several reasons: (i) out of the context, the illustration should mention HBEC specifically; (ii) the representation of the pseudostratified epithelium (even if simplified) should appear more accurately; (iii) the effect on the epithelium height is excessive: it would appear that passaging HBEC induces a decrease of the total epithelium height after differentiation, whereas the authors only show a short decrease of relative cell height during transepithelial water flow measurement.
2. The background must be implemented with additional studies exploring similar biological hypotheses in the context of respiratory research. For example 10.1152/ajpcell.00363.2021 (passaged nasal epithelial cells); 10.1186/s12890-018-0652-2 (asthma vs non-asthma passaged bronchial epithelial cells).
3. Additional information is required in the materials and methods. For instance:
a. line 70: what is the “bronchial epithelial growth media”?
b. line 120: RNAseq experimental conditions should be specified (when were the cells harvested? etc.).
c. line 160: what are the details for surface biotinylation experimental approaches?
4. Regarding the western blotting:
a. The overall quality should be improved.
b. Where is the fourth donor?
c. The differences that are seen are not so convincing.
d. Is it possible to analyze the ratio of apical vs basal?
e. The original full blots should be provided in supplemental data.
f. How did the authors separate apical vs total?
5. The biostatistics regarding gene expression are rudimentary and could be largely implemented with biologically relevant analysis.
Minor comments:
1. In the introduction, the description of mucus-secreting cells is confusing as the vast majority of the mucus present in the airways is produced by the glands and not the surface epithelium.
2. The size of the effective is not always clear. In some experiments, n=6 but only 4 donors were included in the study.
3. Is there a statistical method that could be applied to compare the curves in F1A/C, and F3A/C?
4. GAPDH is not mentioned in western blotting methods while it appears in F2A.
Author Response
Reviewer 2
The manuscript from Tongde Wu et al. reports functional and transcriptomic alterations in passaged primary human bronchial epithelial cells. The findings are important for the research community exploiting airway epithelial cell cultures. The manuscript is well-written but is often brief. The experimental design could have been more ambitious to strengthen the conclusions. It seems better suited as a short communication. In addition, why is the chosen section “cellular immunology”?
[Response] Thanks you for your kind words. We will work with the editors of the journal to find the right publication format and to ensure that the manuscript is contained in the correct section.
Major comment:
- The graphical abstract is misleading for several reasons: (i) out of the context, the illustration should mention HBEC specifically; (ii) the representation of the pseudostratified epithelium (even if simplified) should appear more accurately; (iii) the effect on the epithelium height is excessive: it would appear that passaging HBEC induces a decrease of the total epithelium height after differentiation, whereas the authors only show a short decrease of relative cell height during transepithelial water flow measurement.
[Response] We thank the reviewer for their feedback. In line with their suggestions, the graphical abstract has been updated to represent pseudostratified epithelial without an effect on cell height.
- The background must be implemented with additional studies exploring similar biological hypotheses in the context of respiratory research. For example 10.1152/ajpcell.00363.2021 (passaged nasal epithelial cells); 10.1186/s12890-018-0652-2 (asthma vs non-asthma passaged bronchial epithelial cells).
[Response] We thank the reviewer for highlighting these references. They have been incorporated into the introduction.
- Additional information is required in the materials and methods. For instance:
- line 70: what is the “bronchial epithelial growth media”?
[Response] bronchial epithelial growth media is LHC basal medium [Life Technologies] supplemented with Bovine serum albumin [0.5 mg/mL, Sigma-Aldrich], Bovine pituitary extract [10 μg/mL, Sigma-Aldrich], Insulin [0.87 μM, Sigma-Aldrich], Transferrin [0.125 μM, Sigma-Aldrich], Hydrocortisone [0.21 μM, Sigma-Aldrich], Triiodothyronine [0.01 μM, Sigma-Aldrich],Epinephrine [2.7 μM, Sigma-Aldrich], Epidermal growth factor [25 ng/mL, Invitrogen], Retinoic acid, 5 × 10−8 M, Sigma-Aldrich], Phosphorylethanolamine [0.5 μM, Sigma-Aldrich], Ethanolamine [0.5 μM, Sigma-Aldrich], Zinc sulfate [3.0 μM, Sigma-Aldrich], Penicillin G sulfate [100 U/mL, Sigma-Aldrich], Streptomycin sulfate [100 μg/mL, Sigma-Aldrich], Gentamicina [50 μg/mL, Sigma-Aldrich] and Amphotericin [0.25 μg/mL, Sigma-Aldrich])
This information has been added to the manuscript.
- line 120: RNAseq experimental conditions should be specified (when were the cells harvested? etc.).
[Response] RNA was collected from HBECs after 4 weeks of ALI culture. The manuscript has been updated to include this information.
- line 160: what are the details for surface biotinylation experimental approaches?
[Response] For assessment of apical membrane proteins, apical membrane proteins were biotinylated as previously described20. Polarized HBECs were washed three times with PBS supplemented with 1 mm MgCl2 and 1 mm CaCl2 (PBS2+). Sulpho-NHS-biotin (0.5 mg ml−1) in borate buffer (in mm: 85 NaCl, 4 KCl, 15 Na2B4O7, pH 9) was applied apically and incubated for 30 min with gentle agitation. PBS2+ supplemented with 10% (v/v) FBS was added to the basolateral bath to prevent biotinylation of basolateral proteins. Unbound sulpho-NHS-biotin was quenched with PBS2+ supplemented with 10% (v/v) FBS. Cells were lysed in lysis buffer (0.4% sodium deoxycholate, 1% NP-40, 50 mm EGTA, 10 mm Tris-Cl, pH 7.4 and protease inhibitor (Roche) and protein concentration was determined by bicinchoninic acid assay. Three hundred micrograms of total protein per sample was incubated overnight with 100 μl of neutravidin-agarose beads at 4 °C with agitation. Biotinylated proteins bound to beads were washed three times with lysis buffer and eluted in 30 μl of Laemmli buffer supplemented with 10% (v/v) β-mercaptoethanol by first incubating at room temperature for 10 min, followed by heating at 95 °C for another 10 min.
This information has been added to the manuscript.
- Regarding the western blotting:
- The overall quality should be improved.
[Response] (Please also see response to reviewer 1). We appreciate that the quality of the western blots presented here are suboptimal. Due to the use of HBECs in this study and the quantification of endogenous low abundance proteins, clean, high contrast blots are technically challenging. These difficulties are also encountered by others who have published western blots from HBEC cultures (e.g. PMID: 32457197). We believe that in combination with our functional data demonstrating impairments in fluid transport with passage, that these blots highlight an important impact of passage on the expression of key ion channels, frequently investigated through the use of HBEC cultures. We hope that the quality of our western blots can be re-evaluated in the context of the technical challenges of the culture model used. In line with the reviewer’s comments, we have expanded the blots to display more of the lanes being quantified. Size markers have also been added and the summary data has be updated to remove normalization to passage 0 for each donor.
- Where is the fourth donor?
[Response] We apologize for this error. The study was conducted on cells from 3 donors throughout. The manuscript has been updated to reflect this.
- The differences that are seen are not so convincing.
[Response] We thank the reviewer for their comment. To improve the presentation of our data, we have removed normalization to passage 0 and expanded the field of view of our blots. Also, the changes in CFTR protein that we observed are consistent with the ASL height data.
- Is it possible to analyze the ratio of apical vs basal?
[Response] We thank the reviewer for their suggestion. Unfortunately, our protein quantification only focused on total protein and apical expression measured by surface biotinylating and Co-IP. We therefore do not have data for basolateral expression with which to analyze the ratio of apical to basolateral expression.
- The original full blots should be provided in supplemental data.
[Response] We have updated the western blotting figure to show more of the blot
- How did the authors separate apical vs total?
[Response] For apical studies, apically located proteins were identified by surface biotinylating and Co-IP. Total protein was measured by lysing the cells and running a standard western blot protocol. We have updated the methods to better explain these protocols.
- The biostatistics regarding gene expression are rudimentary and could be largely implemented with biologically relevant analysis.
[Response] We thank the reviewer for their suggestion. We have updated the RNAseq analysis to investigate gene set enrichment in P2 HBECs. Genes sets based on biological function were used (c5.go.bp.v2022.1.Hs.symbols.gmt database of GO Biological Process ontology gene sets) to maximize biological relevance. This approach improves on our previous report as GSEA takes into account all expression changes within a gene set, not just those which were significantly changed. Leading edge analysis was then used to identify key genes driving our predicted phenotypic changes. The methods, figure and table have be undated accordingly.
Minor comments:
- In the introduction, the description of mucus-secreting cells is confusing as the vast majority of the mucus present in the airways is produced by the glands and not the surface epithelium.
[Response] The reviewer correctly highlights the importance of submucosal glands in the production and release of mucus in the large airways. We have updated the introduction to include reference to submucosal glands for clarity.
- The size of the effective is not always clear. In some experiments, n=6 but only 4 donors were included in the study.
[Response] In our original manuscript, n = 6 came from 3 donors with 2 cultures each. To improve transparency, we have amended the graphical data and statistics to reflect 1 data point per donor throughout, with the exception of our RNA seq analysis which remains n = 6 from 3 donors.
- Is there a statistical method that could be applied to compare the curves in F1A/C, and F3A/C?
[Response] We thank the reviewer for their suggestion. We have not curve fitted for Figs 1C, 3A and 3C using linear and single phase exponential equations as appropriate. In all cases, using the F test, we found that the curves were significantly different. We have now redrawn the figures to show the curve fitting and modified the text to reflect this.
- GAPDH is not mentioned in western blotting methods while it appears in F2A.
[Response] we thank the reviewer for highlighting this discrepancy in our methods. They have been updated to include GAPDH.
Round 2
Reviewer 2 Report
The authors responded satisfactorily to my comments. I have no additional remarks.